# The Minimal Residual Disease Using Liquid Biopsies in Hematological Malignancies

**DOI:** 10.3390/cancers14051310

**Published:** 2022-03-03

**Authors:** Rafael Colmenares, Noemí Álvarez, Santiago Barrio, Joaquín Martínez-López, Rosa Ayala

**Affiliations:** 1Hematology Department, Hospital Universitario 12 de Octubre, Instituto de Investigación Sanitaria Imas12, 28041 Madrid, Spain; rafael.colmenares@salud.madrid.org (R.C.); noalva01@ucm.es (N.Á.); santiago_barrio@altumsequencing.com (S.B.); jmarti01@med.ucm.es (J.M.-L.); 2Hematological Malignancies Clinical Research Unit, CNIO, 28029 Madrid, Spain; 3Department of Medicine, Complutense University of Madrid, 28040 Madrid, Spain; 4Centro de Investigación Biomédica en Red de Cáncer (CIBERONC), Instituto Carlos III, 28029 Madrid, Spain

**Keywords:** cell-free DNA, liquid biopsy, cancer, next-generation sequencing (NGS), minimal residual disease, measurable residual disease, molecular residual disease (MRD), leukemia, lymphoma, myeloma, myeloproliferative neoplasms, myelodysplastic syndrome

## Abstract

**Simple Summary:**

Monitoring the response to treatment in hematologic malignancies is essential in defining the best way to optimize patient management. In general, achieving a deeper response has been shown to lead to a better prognosis, and the techniques used to study the minimal residual disease (MRD) are becoming more precise. The use of liquid biopsies, that is, analyzing the presence of alterations in nucleic acids, usually in peripheral blood or other biological fluids, is being studied and optimized with increasingly innovative molecular techniques, such as next-generation sequencing (NGS) in the monitoring of the MRD, avoiding, in many cases, more invasive tests in different hematological neoplasms. Currently, liquid biopsies are not standardized for the MRD monitoring, but there is increasing evidence of its correlation with other techniques to measure responses to treatments and patient outcomes.

**Abstract:**

The study of cell-free DNA (cfDNA) and other peripheral blood components (known as “liquid biopsies”) is promising, and has been investigated especially in solid tumors. Nevertheless, it is increasingly showing a greater utility in the diagnosis, prognosis, and response to treatment of hematological malignancies; in the future, it could prevent invasive techniques, such as bone marrow (BM) biopsies. Most of the studies about this topic have focused on B-cell lymphoid malignancies; some of them have shown that cfDNA can be used as a novel way for the diagnosis and minimal residual monitoring of B-cell lymphomas, using techniques such as next-generation sequencing (NGS). In myelodysplastic syndromes, multiple myeloma, or chronic lymphocytic leukemia, liquid biopsies may allow for an interesting genomic representation of the tumor clones affecting different lesions (spatial heterogeneity). In acute leukemias, it can be helpful in the monitoring of the early treatment response and the prediction of treatment failure. In chronic lymphocytic leukemia, the evaluation of cfDNA permits the definition of clonal evolution and drug resistance in real time. However, there are limitations, such as the difficulty in obtaining sufficient circulating tumor DNA for achieving a high sensitivity to assess the minimal residual disease, or the lack of standardization of the method, and clinical studies, to confirm its prognostic impact. This review focuses on the clinical applications of cfDNA on the minimal residual disease in hematological malignancies.

## 1. Introduction

Hematological malignancies are the result of molecular alterations that affect the genes involved in cell growth and proliferation. Sometimes, the molecular profile of a tumor is obtained only by analyzing a small portion, as in lymphomas, which can lead to the loss of information because of the heterogeneity of the tumor or the presence of multiple tumor sites. In addition, it may, sometimes, be difficult to obtain a good sample to analyze, which involves invasive procedures and increases the risk to the patient. Monitoring the tumor by repeat biopsies is usually not feasible [1].

### 1.1. Liquid Biopsy Components

Various studies have shown that the blood contains remnants of some tissues, including tumor tissues. The term “liquid biopsy” is an attempt to approximate the molecular profile of a tumor by analyzing the peripheral blood using different methods: circulating tumor cells (CTCs), cell-free circulating nucleic acids (DNA, mRNA, micro-RNA, or non-coding RNA), “tumor-educated platelets” (TEPs), or exosomes [2,3].

Circulating tumor cells (CTCs): CTCs come from tumors as an early step in blood-borne metastasis. CTCs are transient in blood, with a half-life of 1–2.4 h and are presented in a low abundance in most patients. Various techniques have been defined to isolate and analyze CTCs. These cells can even be used to establish cell line models to carry out therapeutic studies [4]. In hematological malignancies, the analysis of CTCs is possible in acute myeloid leukemia (AML) or acute lymphoblastic leukemia (ALL), myelodysplastic syndromes (MDSs), myeloproliferative neoplasms (MPNs), multiple myelomas (MM), and some lymphomas, such as mantle cell lymphoma (MCL), follicular lymphoma (FL), marginal zone lymphoma, small lymphocytic lymphoma, and a subset of Burkitt lymphoma. In contrast, diffuse large B-cell lymphoma (DLBCL) and classical Hodgkin lymphoma (cHL) do not typically harbor CTCs [5];RNA: MicroRNAs (miRNAs) are a class of small molecules of 19–24 nucleotides in length, and they are the most abundant RNA molecules in the blood; they can be carried in the exosomes or TEPs. They have a high stability and play an important role in tumor growth and treatment resistance [6];Tumor-educated platelets (TEP): Platelets are circulating anucleated fragments originating from megakaryocytes in the bone marrow, and they participate in hemostasis and the initiation of wound healing. However, they also have a role in systemic and local responses to tumor growth, as tumor cells alter the RNA profile of these platelets. In addition, TEPs can ingest the circulating mRNA released by tumor cells or solubilized tumor-associated proteins [7]. These interactions may signify a potential for cancer diagnosis or monitoring [8];Exosomes: Exosomes are a type of extracellular vesicle of endocytic origin, ranging in size between 30 and 100 nm; they are detectable in the blood of patients with some types of cancer, and they carry proteins and nucleic acids. They are analyzed through their RNA content [9]. For example, the ability of HL-specific exosomal microRNAs (miRNAs) to inform a treatment response in HL has been studied. The authors found that the specificity and sensitivity of exosomal miRNAs are superior to protein-bound miRNAs, with regard to HL detection [10];DNA: The existence of cell-free DNA (cfDNA) was first described in 1948, and some studies have reported a higher amount of cfDNA in patients with cancer [11,12,13]. Initially, some reports have identified apoptosis, necrosis, or both, as the main source of cfDNA. During apoptosis, the chromosomes are trimmed by DNases into multiple nucleosomal units of 180 bp that are released into the blood stream. These fragments “circulate” for one to three hours before they are ingested by phagocytes and other cells, whereas the cfDNA is completely digested into nucleotides by lysosomes [14]. Necrosis causes the random nonspecific and incomplete digestion of DNA. Some studies have suggested that DNA may be released by living cells [15,16,17]. This review is focused on circulating tumor DNA (ctDNA) and cfDNA, and there are differences between these concepts. cfDNA fragments are longer, with a size range of 160–180 bp, indicating a caspase-dependent apoptotic cleavage, while ctDNA fragments range between 90 and 150 bp, and others range between 250 and 320 bp [18]. The proportion of ctDNA in cfDNA varies, and the most-used methods to detect cfDNA are the real-time quantitative polymerase chain reaction (qRT-PCR) and next-generation sequencing (NGS) [19,20]. Liquid biopsies, by the quantification of cfDNA, can be used for all types of lymphomas, because the plasma from peripheral blood (PB) regularly contains low levels of detectable lymphoma-derived ctDNA, as well as in myeloid pathology (AML, MDS, and MPN) or other lymphoid pathology (MM and ALL) [21]. Therefore, the ctDNA-based liquid biopsy has emerged as a platform to genotype these tumors;

The first methods used to analyze ctDNA were based on PCR, using technologies such as TaqMan, PNA clamps, and the Scorpion Amplification Refractory Mutation System; however, they were limited because of their analytical sensitivity and specificity [22]. In the last decade, other newer techniques have been used: digital PCR (dPCR) and NGS have emerged, increasing the detection thresholds by up to 0.01% for mutant allele abundance [23];

○dPCR-based methods have a high sensitivity (0.01%) (better than real-time quantitative PCR), but they can only detect a few alterations simultaneously, and they must be optimized for each mutation. In this technique, thousands of individual reactions are analyzed with each sample, calculating whether a PCR end-point is reached, and evaluating the absolute number of molecules in the sample. It is useful in liquid biopsies because it is a precise and sensitive technique, detecting low-abundance targets. [24];○In NGS, millions of DNA fragments are generated in a single sequencing process; enrichment can be performed to select the areas of interest. Finally, a massive and parallel sequencing is carried out. NGS-based methods can detect multiple alterations simultaneously. Initially, they had an insufficient sensitivity (>1%), but various groups have been developing methods that allow for a greater sensitivity. Indeed, some of them have tried to use the deep sequencing of a limited number of amplicons for the commonly mutated genes in cancer. The technique, called CAncer Personalized Profiling by deep Sequencing (CAPP-Seq), can detect all major classes of mutations (single nucleotide variants, indels, rearrangements, and copy number alterations). Capture-based NGS methods enrich genomic regions, before sequencing, by a hybridization of target regions to antisense oligonucleotides. With this technique, large portions of the genome can be examined [25].

### 1.2. Minimal Residual Disease Using Liquid Biopsy

The minimal/measurable residual disease (MRD) is defined as the persistence of a small number of malignant cells after the initial treatment, undetectable by morphologic or conventional screening methods. Table 1 and Table 2 detail the methods used in clinical practice to evaluate the response to treatment in different hematological malignancies. 

In myeloid pathology, the MRD-guided approaches have become an attractive therapeutic strategy (e.g., CML, AML, and MDS), allowing for a more individualized therapy, possibly leading to less treatment-related toxicities and better outcomes. Moreover, the treatment, upon molecular relapse, is more effective than upon hematologic relapse (e.g., after allogeneic HCT in AML and MDS) [26]. The monitoring of the clonal evolution of AML identifies the leukemia subtype, clinical outcome, and potential new drug targets for post-remission strategies or relapse [27,28]. However, a universal MRD marker for MDS and AML is unlikely, because of the genotypic and phenotypic heterogeneity of these diseases [29]. Thus, a more effective strategy may be individualized MRD monitoring using a targeted next-generation sequencing panel [30].

In lymphoid pathology, the MRD-guided approaches, mainly based on the monitoring of the clonal rearrangements of Igs or TCR, have also become a therapeutic strategy for ALL and MM. However, the available MRD strategies have not yet been incorporated into the assistance practice for other lymphoid tumors, where the imaging methods are the method of choice to assess responses to treatments.

Liquid biopsies have several advantages over bone marrow for the detection of MRD: They are less invasive, they may provide a more comprehensive molecular overview of tumor heterogeneity, and they make it easier to obtain repeated blood samples over time to understand the dynamics of the response to a specific treatment. Multiple studies have demonstrated a satisfactory positive predictive value of ctDNA detection, as the presence of the MRD identified through ctDNA is associated with a lower disease-free survival (DFS) rate. Nevertheless, several hurdles maintain the negative predictive value of the technique at a low level, namely, the detection methods are not sensitive enough to detect ctDNA at very low amounts, new clones that are not captured by the selected technique can emerge, and the timing of the post-treatment sampling can be inappropriate [31]. Moreover, in the case of cfDNA, the logistics of the sample processing are crucial to avoid white blood cell (WBC) lysis and the subsequent genomic DNA contamination. To avoid this issue, samples need to be processed in less than four hours, or collected in tubes with a cell stabilizer [32].

**Table 1 cancers-14-01310-t001:** Minimal residual disease monitoring methods in myeloid malignancies.

Method	APL	AML	MDS	MPN	CML
Image Methods	No	No	No	Yes	No
CT or PET/CT MRI				Spleen measurement by CT or MRI (clinical trials)	
Histologic/morphologic methods	Yes, BM or PB, 10^−2^	Yes, BM, 10^−2^	Yes, BM, 10^−2^	Yes, BM, 10^−2^ (clinical trials)	No
MFC methods	No	Yes, BM, 10^−4^ or 10^−5^	Yes, BM, 10^−4^ or 10^−5^	No recommendations	No
Molecular methods	Yes, RQ-PCR (*PML/RARA*), BM, 10^−5^	RQ-PCR *, BM or PB, 10^−6^	No recommendations	No recommendations	Yes, PB (*BCR/ABL1*), 10^−5^
NGS methods	No	NGS **, BM, 10^−6^	Investigational use (clinical trials)	Investigational use (clinical trials)	No
Timing of MRD assessment	Post-induction time and PCR every 3 m for 2 years	Upon completion of the initial induction, additional time points should be guided according to the regimen used before allogeneic transplantation	No recommendations	Only in clinical trials ***	PCR every 3 m for one year, then every 6 m
References	[33,34]	[34]	[35]	[36,37,38]	[39]

APL, acute promyelocytic leukemia; AML, acute myeloid leukemia; MDS, myelodysplastic syndrome; MPS, myeloproliferative syndrome; CML, chronic myeloid leukemia; CT, computed tomography; PET, positron emission tomography; MRI, magnetic resonance imaging; CR, complete response; y, year; BM, bone marrow; PB, peripheral blood; FISH, fluorescence in situ hybridization; MFC, multiparametric flow cytometry; RQ-PCR, real quantitative PCR; ASO-PCR, allelic specific oligonucleotide PCR; NGS, next-generation sequencing; MRD, minimal residual disease; m, month. The information in the boxes includes the test used, the sample to be studied, and the sensitivity of the method. * *CBFb-MYH11*, *RUNX1-RUNX1T1*, and mut*NPM1*. ** Excluded DTA mutations. *** 2013 IWG-MRT and ELN guidelines recommend monitoring the response (anemia response, spleen response, and symptom response), as well as signs and symptoms of disease progression every three to six months during the course of treatment. BM should be performed as clinically indicated.

**Table 2 cancers-14-01310-t002:** Minimal residual disease monitoring methods in lymphoid malignancies.

Method	ALL	DLBCL	FL	HL	CLL	MM
Image Methods	No	Yes	Yes	Yes	Yes	Yes
CT or PET/CT MRI		PET-TAC scan or CT scan contrast	PET-TAC scan or CT scan contrast	CR includes PET negative within 3 m posttreatment. Consider body CT with contrast no more often than every 6 m for the first 2 y following completion of therapy	Lymphoid nodes, spleen, and liver evaluation by CT	PET/CT
Histologic/morphologic methods	Yes, BM, 10^−2^	BM biopsy (optional)	BM biopsy (optional)	No, unless there is BM involvement at diagnosis	Yes, BM, 10^−2^	Yes, BM with FISH, 10^−2^
MFC methods	Yes, BM, 10^−4^	No	No	No	Yes, BM or PB, 10^−5^	Yes, BM, 10^−5^
Molecular methods	RQ-PCR, BM, 10^−6^	No	No	No	ASO-PCR, BM or PB, 10^−4^	No
NGS methods	NGS Igs-TCR, BM, 10^−6^	NGS liquid biopsy, investigational use	NGS liquid biopsy, investigational use	Investigational use	NGS Igs, BM or PB, 10^−6^	NGS Igs, BM, 10^−6^
Timing of MRD assessment	Upon completion of initial induction and additional time points; should be guided according to the regimen used	Post-third cycle and every 3 m	Post-third cycle every 3–6 m for 5 years	Post-third cycle. Additional time points should be guided according to the regimen used	Post-third cycle every 3–6 m for 5 years	Post-third cycle. Consider body CT with contrast no more often than every 6 m for the first 2 y following completion of therapy
References	[40]	[41]	[41]	[42]	[43]	[44]

ALL, acute lymphoblastic leukemia; DLBCL, diffuse large B-cell lymphoma; FL, follicular lymphoma; HL, Hodgkin lymphoma; CLL, chronic lymphatic leukemia; MM, multiple myeloma; CT, computed tomography; PET, positron emission tomography; MRI, magnetic resonance imaging; CR, complete response; y, year; BM, bone marrow; PB, peripheral blood; FISH, fluorescence in situ hybridization; MFC, multiparametric flow cytometry; RQ-PCR, real quantitative PCR; ASO-PCR, allelic specific oligonucleotide PCR; NGS, next-generation sequencing; MRD, minimal residual disease; m, month; y; year. The information in the boxes includes the test used, the sample to be studied, and the sensitivity of the method.

## 2. Methods

References for this review were identified through searches of PUBMED with the search terms “cell free DNA,” “liquid biopsy,” “MDS,” “MPN,” “AML,” “NHL,” “CLL,” “MM,” and “ALL,” published from 1 January 2000 to 31 December 2021. Articles were also identified through searches of the reference authors’ names. Only papers published in English were reviewed. The final reference list was generated on the basis of significance and relevance to the broad scope of this review. Note that when we searched for the term cfDNA and cancer, we found 20,446 papers; however, if we associated these terms with MRD, only 116 results appeared. If we carried out this specific search for leukemia, lymphomas, or myelomas, four, seven, or seven results appeared, respectively. This shows that there are still not many publications with the relevant data on the usefulness of liquid biopsies to assess the MRD. However, there is a great interest in this application and much work is being conducted on different approaches in order to optimize these liquid biopsy techniques to achieve an adequate sensitivity for detecting tumor persistence or relapse.

## 3. Myeloid Malignancies

### 3.1. Acute Myeloid Leukemia

Acute myeloid leukemia (AML) is a malignancy of the hematopoietic stem cell precursors of the myeloid lineage, causing an overproduction of neoplastic clonal myeloid stem cells and the infiltration of extramedullary organs [45,46,47,48,49]. A correct diagnosis and classification of the disease are critical for the treatment of patients with AML, and an evaluation of the response to the treatment by the MRD quantification is essential to optimize treatment and select the appropriate method of response consolidation. The quantification of the MRD is very important because it can identify those patients at a high risk of relapse. Currently, studies including the morphologic evaluation on bone marrow (BM) aspirate or biopsies, cytogenetic analyses, and molecular testing using gene panels or NGS are currently being carried out for diagnoses and follow-ups [50]. Nevertheless, several studies have explored the use of peripheral blood (PB) as an input for the detection of residual disease, and the MRD obtained from the PB is already being incorporated into AML protocols for clinical decision making (e.g., *CBF* and *NPM1* AML Pethema protocols with the MRD by RT-qPCR for follow-up *RUNX1-RUNX1T1*, *CBFb-MYH11*, and *NPM1* mutations) [51,52,53]. Maurillo et al. analyzed BM and PB samples in 50 AML patients, 48 of whom were re-evaluated after consolidation. The preliminary results showed a concordance in the CTC quantification between PB and BM samples, increasing the disease stratification value and improving the MRD monitoring in AML patients. The analysis of the MRD after consolidation revealed useful prognostic information [54]. 

The MRD in AML is defined as the presence of leukemic blasts at a level lower than the limit of conventional morphologic detection (1:1000–1:10^6^ white blood cells) [55]. The main techniques used, thus far, for the MRD in AML patients are multiparameter flow cytometry (MFC) and molecular techniques using RT-qPCR. The detection of the MRD by MFC uses the leukemia-associated immunophenotypes (LAIPs) approach (LAIPs are defined at diagnosis and their presence is subsequently monitored at follow-up) and the DfN approach (DfN, different from normal, irrespective of the LAIPs at diagnosis) and is applicable in most AML patients, with a sensitivity reported from 10^−3^ to 10^−5^. The molecular MRD is more sensitive and specific than flow cytometry, and real-time quantitative PCR (RT-qPCR) can achieve a sensitivity of 10^−4^–10^−6^ [51,56,57,58,59]. Droplet digital PCR (ddPCR) can reach a sensitivity of 10^−4^–10^−6^ as well [52,58], and NGS can achieve a sensitivity of 10^−4^–10^−5^ [27,60,61]. The problem with RT-qPCR is that that its applicability is limited to 30–40% of AMLs. NGS has an acceptable sensitivity, but with a better applicability than RT-qPCR (>85% of AML cases), and it provides new insights into the evolution of subclones that emerge under treatment, or during disease progression. This allows the availability of markers that can be used to measure the MRD to be expanded [62,63,64]. 

Advances in NGS technology have also enabled the sequencing of increasingly low amounts of DNA, using a variety of sources [65]. Blood contains many types of biological materials, such as circulating cells, mRNA, proteins, and cfDNA. In the blood of patients with AML, a part of the ctDNA is released by the tumor [66], but the tumor cells themselves circulate in the peripheral blood of these AML patients, and are called blastic cells. This CTC quantification has been incorporated into protocols, as previously cited. 

However, there is not much reported experience with cfDNA in AML patients (Table 3). Through new technologies, the number of studies carried out in this field is increasing. Short et al. analyzed cfDNA and BM samples in 22 AML patients. They analyzed 28 genes and found that five mutations were detected only in the cfDNA, 15 mutations were detected in the BM, and 19 mutations were detected in both samples. Generally, the mutations detected in only one source had a variant allele frequency (VAF) of less than 10%. This suggests that both methods could miss subclonal populations. In addition, the cfDNA samples detected new or persistent mutations that implied a relapse. The results suggest that cfDNA and BM are complementary in the follow-up and monitoring of diseases in these patients [67].

Nakamura et al. conducted a study in 51 patients (37 with AML and 14 with MDS) to investigate the role of ctDNA in the risk of relapse after undergoing myeloablative allogeneic hematopoietic stem cell transplantation (alloSCT), with a median follow-up of 32 months post-alloSCT, where 16 out of 51 patients relapsed at a median of seven months. They collected ctDNA and BM samples at diagnosis and at follow-up. NGS studies with a 54- or 141-gene panel, or WES, were carried out. They were able to observe an increase in ctDNA in relapsed patients, determining that non-invasive testing has comparable utility in the BM testing, predicting the risk of relapse [68]. In this work, *DNMT3A*, *TET2*, and *ASXL1* mutations (DTA gene mutations) were prognostic factors of leukemia relapse.

Rausch et al. demonstrated the application of a new method called the double drop-off digital droplet PCR (DDO-ddPCR) for the detection of gene mutations in *NPM1*, *IDH2*, and *NRAS*, which can detect and quantify diverse alterations at two nearby hotspot regions present in these genes. They used this method for disease monitoring, and compared it to qPCR, ddPCR, and NGS. At 38 time points (78%), the results of the cfDNA-ddPCR and BM qPCR were concordant, whereas at 11 time points (22%), during the follow-up, an *NPM1* mutation was detected in the BM but not in the PB cfDNA. The cfDNA analysis was found to have a similar sensitivity compared to the quantitative PCR-based analysis of peripheral blood, but a lower sensitivity compared to the BM qPCR [69]. 

In conclusion, the use of liquid biopsies in AML shows promising results in terms of disease staging, monitoring, prognosis, and treatment [67], as well as in the detection of early relapse [70]. Although CTCs and cfDNA have been shown to be new biomarkers in the blood [25], at the moment, these cannot yet replace BM studies, mainly due to the low concentration of cfDNA in the blood [67]. An additional obstacle in the current molecular MRD approaches, and one that also affects liquid biopsy studies, is to find the molecular variants that represent, and are specific to, leukemic cells, with the capability of relapsing [71]. The determination of the MRD is complicated by the fact that many treated patients have persistent clonal hematopoiesis (CH) that may not reflect residual AML [72]. For this, the ELN group in the consensus document for the MRD in AML 2021 provided the following recommendations: To consider all the detected mutations as potential MRD markers; germline mutations (VAF of ~50% in the genes *ANKRD26*, *CEBPA*, *DDX41*, *ETV6*, *GATA2*, *RUNX1*, and *TP53*) should be excluded as NGS-MRD markers, as they are non-informative for the MRD; and DTA mutations can be found in age-related clonal hematopoiesis and should be excluded from the MRD analysis, as the mutations associated with clonal hematopoiesis often persist during remission and, thus, may not represent the leukemic clone. If the only detectable mutations are in the DTA genes, it is recommended to use MFC and/or PCR for the MRD assessment; mutations in signaling pathway genes (e.g., *FLT3-ITD*, *FLT3-TKD*, *KIT*, *KRAS*, and *NRAS*) likely represent residual AML when detected, but they are often subclonal and have a low negative predictive value. These mutations are best used in combination with additional MRD markers; NGS-MRD analyses in patients treated with targeted agents (*FLT3* and *IDH1/IDH2* inhibitors) should include the molecular marker that is targeted, but also others that are present in the sample [33].

**Table 3 cancers-14-01310-t003:** Liquid biopsies in acute myeloid leukemia.

Target	Methods	Cohort Size/Disease Stage	Evidence: Key Points	Application	Reference
*IDH1* and *IDH2* genes (R140 and R172 mutations)	Sanger, ddPCR, NGS, and qPCR	*n* = 60 Diagnosis	Give evidence that the drop-off ddPCR is a valid new molecular tool for detecting IDH2 mutations Techniques such as ddPCR, NGS, and qPCR started to be implemented	MRD	Grassi et al., 2020 [53]
*CEBPA* mutations and blast cells	RT-qPCR	*n* = 4 (*n* = 3 diagnostic, *n* = 1 relapse)	RT-qPCR can achieve a sensitivity of 10^−4^–10^−6^ Describe specific RT-qPCR for CEBPA mutations	Concordance	Smith et al., 2006 [56]
*NPM1* mutations	Flow cytometry, RT-qPCR	*n* = 15 patients *n* = 45 MRD samples Diagnosis and follow-up	Knowledge of RT-qPCR-based MRD results RT-qPCR has a higher sensitivity than FC (10^−^^4^–10^−^^6^ vs. 10^−3^–10^−5^)	MRD	Pettersson et al., 2016 [51]
Somatic mutations	ddPCR, RT-qPCR	*n* = 41 patients with AML-M1/M2 *n* = 20 healthy volunteers	ddPCR can achieve a sensitivity of 10^−4^–10^−6^ Identify genes that contribute to leukemogenesis	Concordance	Handschuh et al., 2017 [52]
Residual leukemic cells	Flow cytometry	*n* = 135 patients with de novo AML (100 achieving CR after intensive chemotherapy)	Flow cytometry achieves a sensitivity of 10^−4^ MRD patients have a five-year RFS >70%, MRD+ patients have the worst prognosis Incorporate MRD assessment in the protocols for the treatment of AML	MRD Response assessment	Buccisano et al., 2006 [55]
Somatic mutation	NGS	*n* = 22 (after remission) post-treatment	cfDNA and BM were complementary in the follow-up and monitoring of disease The concordance of the VAF assessment by both methods was high (*R*^2^ = 0.849)	Concordance MRD	Short et al., 2020 [67]
Driver mutations	NGS dPCR	*n* = 53 (*n* = 37 AML, *n* = 14 MDS) (after post-alloSCT) Diagnosis and post-treatment	An increase in ctDNA in relapsed patients ctDNA has concordance with BM testing, having a comparable utility	Concordance MRD Response assessment	Nakamura et al., 2019 [68]
Somatic mutation	DDO-ddPCR dPCR qPCR NGS	*n* = 57 samples (cfDNA), *n* = 28 (PB), *n* = 53 (BM) Post-treatment	Demonstrate the application of DDO-ddPCR in comparison to dPCR and qPCR DDO-ddPCR has a sensitivity of 0.037% cfDNA opens new strategies for response assessment, disease monitoring, and molecular profiling of MRD	Concordance MRD	Rausch et al., 2021 [69]
Leukemic cells	Flow cytometry	*n* = 50 patients with de novo AML	BM and PB samples were significantly concordant (*r* = 0.86 and 0.82, respectively, *p* < 0.001) The cut-off value of residual leukemic cells was 1.5 × 10^−4^ PB MRD was found to have a significant effect on relapse-free survival (*p* = 0.036).	Concordance MRD	Maurillo et al., 2007 [54]
Primitive blast (CD34+/CD117+/CD133+)	Flow cytometry	*n* = 114 patients (205 paired BM and PB samples) Diagnosis and post-treatment	Primitive blast frequency was lower in the PB; PB MRD is more specific than BM The role of MRD in the PB may have an essential use in the future clinical management of AML patients Flow cytometry has a sensitivity of 10^−3^–10^−5^	Concordance MRD Response assessment	Zeijlemaker et al., 2016 [59]

Ref., reference; ddPCR, droplet digital PCR; NGS, next-generation sequencing; qPCR, quantitative PCR; MRD, minimal residual disease; RT-qPCR, real-time quantitative PCR; FC, flow cytometry; CR, complete response; RFS, relapse-free survival; dPCR, digital PCR; BM, bone marrow; AML, acute myeloid leukemia; MDS, myelodysplastic syndrome; SCT, stem cell transplantation; VAF, variant allele frequency; DDO-ddPCR, D-aspartate oxidase ddPCR, droplet digital PCR; PB, peripheral blood; cfDNA, cell-free DNA.

### 3.2. Myelodysplastic Syndromes

Myelodysplastic syndromes are a heterogeneous group of hemopathies characterized by a clonal disorder of hematopoietic stem cells, and whose main problems are caused by the morbidities associated with cytopenias, as well as by the potential risk of progression to acute myeloid leukemia. It is a disease typical of older patients, and in most cases, it is not possible to opt for treatments with curative options, so it is not a pathology in which the MRD techniques have been developed, as in others (e.g., AML). However, in recent years, several papers have been published highlighting the usefulness of cfDNA monitoring in patients with MDS (Table 4).

Dawson published a work where the detection of mutations and cytogenetic alterations in cfDNA predicts treatment failure. Targeted deep sequencing (TS) was performed on DNA derived from BM and plasma using a customized panel of 55 genes. The sequencing of the BM samples identified putative driver mutations in 10 out of of 12 patients. A digital polymerase chain reaction (dPCR) was performed to validate the TS results. The quantification of the mutant allele fraction (MAF) showed an excellent correlation between TS and dPCR. Through the analysis of serial bone marrow and the matched plasma samples (*n* = 75), the authors demonstrated that ctDNA is directly comparable to bone marrow biopsies in representing the genomic heterogeneity of malignant clones in MDS, and the serial monitoring of ctDNA allows the concurrent tracking of mutations throughout therapy, which could be able to anticipate treatment failure. In addition, in the time points with severe neutropenia, the allelic burden by variant allele frequency (VAF) determination was higher in the plasma cfDNA than leucocyte BM [73]. 

Nakamura published the prognostic impact of the circulating tumor DNA status in a post-allogeneic hematopoietic stem cell transplantation in AML and MDS. Fifty-three patients with MDS/AML were studied by the targeted 37-gene panel NGS at diagnosis, which detected 51 cases (96.2%) with driver mutations, at diagnosis, by NGS with a limit of detection of 0.04%. The follow-up was made by ddPCR in serum samples with one or more markers, with a limit of detection of 0.1–0.01%. They found that ctDNA reflects clonal dynamics, and the persistent molecular MRD status in the post-allogeneic stem cell transplantation (SCT) predicts relapse and survival. Increased ctDNA levels between month 1 and month 3, after the allogeneic transplantation, could be a predictor of relapse. The use of serum ctDNA has a benefit over leukocyte DNA in predicting relapse, with a concordance observed in 8/11 cases; however, in cases with cytopenias, it was only detectable in serum ctDNA [68].

In another study, the peripheral plasma cfDNA samples available from patients with aplastic anemia (AA; *n* = 25), MDS (*n* = 27), and a healthy cohort (*n* = 107) were screened for somatic variants in genes related to the hematologic malignancies by a targeted-panel NGS. The results were further compared to the DNA sequencing of matched blood cells. The authors observed that the concordance between cfDNA and the blood cells was poor for clonal hematopoiesis (CH) detection when variants were at a variant allele frequency <10%, which was mostly observed in the healthy and AA cohort, but not in the MDS group. The use of cfDNA does not offer advantages over CTC for the detection of variants associated with clonal hematopoiesis in diseases with low allele burdens and healthy individuals. They detected numerous new variants with a low number of reads in cfDNA that they considered artifacts, so they recommended strict filtering criteria in these cases. Ultra-sensitive assays with a robust sequencing coverage and error-correction methodology may be required to overcome assay discordance [74].

**Table 4 cancers-14-01310-t004:** Liquid biopsies in myelodysplastic syndromes.

Target	Methods	Cohort Size/Disease Stage	Evidence: Key Points	Application	Ref.
Somatic mutations	NGS (55 genes)	*n* = 12 patients 75 samples at follow-up	ctDNA is directly comparable to a BM biopsy in representing the genomic heterogeneity of malignant clones in MDS Serial monitoring of ctDNA allows concurrent tracking of mutations and could be able to anticipate treatment failure	MRD Concordance	Yeh et al., 2017 [73]
Somatic mutations	NGS diagnostic (37 genes) ddPCR for follow-up	*n* = 51 (*n* = 15 novo AML, *n* = 22 secondary AML, *n* = 14 MDS) LOD (NGS) = 0.04% LOD (ddPCR) = 0.1–0.01%	ctDNA reflects clonal dynamics, and persistent molecular MRD post-alloSCT predicts relapse and survival, with the usefulness of DTA mutations Relevance of ctDNA over bulk PB analysis, especially when there are cytopenias	Concordance MRD	Nakamura et al., 2019 [68]
Somatic mutations (CHIP quantification)	NGS (200 genes)	*n* = 25 AA patients, *n* = 27 MDS patients, *n* = 107 healthy controls LOD = 0.96%	The concordance between cfDNA and blood cells was poor for clonal hematopoiesis detection when variants were at a VAF < 10%, and stringent criteria to filter out discordant variants improved cfDNA concordance with blood cells Quantification of CHIP in cfDNA was not comparable to blood cells	Concordance	Gutierrez-Rodrigues et al., 2021 [74]

Ref., reference; NGS, next-generation sequencing; ctDNA, circulant tumor DNA; BM, bone marrow; MRD, minimal residual disease; ddPCR, droplet digital PCR; AML, acute myeloid leukemia; MDS, myelodysplastic syndrome; LOD, limit of detection; alloSCT, allogeneic stem cell transplantation; PB, peripheral blood; CHIP, clonal hematopoiesis of indeterminate potential; AA, aplastic anemia; cfDNA, cell-free DNA.

### 3.3. Myeloproliferative Neoplasms 

Myeloproliferative neoplasms are disorders characterized by stem cell-derived clonal myeloproliferation, with mutually exclusive *JAK2*, *CALR*, and *MPL* mutations as phenotype-driver variants [75,76]. In myeloproliferative neoplasms, the aim of the available treatments is to avoid thrombosis. Good treatments that decrease the disease burden or cure the disease are not available, with the exception of allogeneic transplantation in advanced stage primary myelofibrosis (PMF). To date, drug therapy has not been shown to improve survival or prevent leukemic/fibrotic transformation in either essential thrombocythemia (ET) or polycythemia vera (PV); therefore, treatment is primarily directed at preventing thrombotic complications [77]. 

Based on these assumptions, the usefulness of markers to quantify residual disease is unclear (Table 5). However, Bellosillo et al. [78] published a work where 107 patients with MPNs were studied by NGS and ddPCR, including 33 PV, 56 ET, 14 PMF, and four unclassifiable MPNs (uMPNs). The aim of the study was to assess the accuracy and reliability of cfDNA analyses in a cohort of MPN patients, comparing this technique to the genotype of peripheral blood granulocytes. The authors showed that the amount of cfDNA in plasma varies among MPN disease phenotypes. A significant positive correlation was observed between the amount of cfDNA and PMF (vs. PV), the leukocyte count, LDH, *MPL*-mutated group, and those suffering from a thrombotic event at the time of diagnosis or during follow-up. In addition, cfDNA and granulocyte DNA showed an equivalent mutational profile, and the discrepancies were detected with variants at low VAF. MPL-mutated and *ASLX1*-mutated patients had higher amounts of cfDNA. However, in relation to the role of cfDNA as a way of monitoring responses to treatment, for the PV cases, in those who received hydroxycarbamide, the *JAK2V617F* VAF remained stable in both the granulocytes and cfDNA during the follow-up. For the ET cases, those who were treated with interferon had a proportional decrease in the *JAK2V617F* VAF, which was observed in granulocytes and cfDNA [78].

## 4. Lymphoid Malignancies

### 4.1. Acute Lymphoblastic Leukemia 

Acute lymphoblastic leukemia (ALL) is a disease with an incidence of one to five cases per 100,000 people in the population. Of these, 67% are B-ALL, and 75% are produced in children under six years of age. The prognosis of the disease is defined by age, the leukocyte count, genetic and molecular alterations, and initial responses to treatment, among others [79]. Since this disease affects the pediatric population, it is even more important to avoid invasive tests, so liquid biopsies would represent an important advance. 

In recent years, a management that takes into account the result of the MRD has been imposed, such as in adults who may require allogeneic hematopoietic stem cell transplantations. For this, the most commonly used methods, to date, are the multiparametric flow cytometry- (more standardized) and PCR-based tests, such as in ALL with *BCR/ABL1* mutations. NGS is not yet routinely used for the MRD testing [80,81,82]. However, guidelines (e.g., NCCN) already recommend the use of NGS to detect *IGH* and *TCR* rearrangements as a marker for the MRD. A comparison of the MRD levels in ALL patients, detected by IG/TR RT-qPCR vs. *BCR-ABL1* genomic transcript quantification, showed a good correlation, and the sensitivity significantly increased when large numbers of cells were acquired [83]. In adult patients with ALL undergoing cellular therapies (hematopoietic cell transplantation and chimeric antigen receptor T-cell therapies), the authors demonstrated a strong concordance between the NGS-based MRD detected in the PB and BM (*r* = 0.87; *p* = 0.001), with a sensitivity and specificity of the MRD detection in the PB of 87% and 90%, respectively, relative to the MRD in the BM [84].

The use of liquid biopsies (cfDNA) in the MRD in ALL has been explored in some studies (Table 6). Van der Velden et al. studied Ig/TCR rearrangements in T-ALL, and its precursor in bone marrow and PB samples, finding a strong correlation in the MRD levels in T-ALL, but not in precursor, B-ALL [85]. Schwarz et al. conducted one of the first studies to see that there was more plasma DNA in patients with ALL than in healthy patients [86]. Cheng et al. compared flow cytometry in bone marrow and peripheral blood plasma PCR quantifying *Ig/TCR* gene rearrangements. They did not show a correlation, but the peripheral blood could predict a relapse [87]. Another study showed that *BCR/ABL1*-positive ALL patients had lower pre-transplant cfDNA levels than *BCR/ABL1*-negative ALL patients [88]. More studies are needed to use cfDNA for monitoring the MRD in ALL.

### 4.2. Lymphomas and Chronic Lymphocytic Leukemia 

Lymphomas are the most common hematologic malignancies, and the gold standard for their diagnosis is tissue or lymph node biopsies, with a pathological examination after surgical resection or lymph node puncture; however, these are invasive examinations (sometimes, as in CNS (central nervous system) lymphoma, the sample may be not accessible) and a biopsy is not useful for monitoring the disease [89]. It is a heterogenous group of neoplasms, with significant differences in the treatment schemes, and non-Hodgkin lymphoma is the most frequent type [90].

Traditionally, imaging scans (including computed tomography (CT) and positron emission tomography (PET)) have been used to assess the responses to treatment of lymphomas, which have several limitations, such as lack of tumor specificity, an inability to detect the disease at the molecular level, an inability to capture dynamic tumor processes, and a radiation risk or cost. However, the study of cfDNA may have advantages (Table 7): A high tumor specificity, a high sensitivity, broad applicability, and the ability to assess tumor heterogeneity [91]. The study of cfDNA in lymphomas includes cfDNA concentrations, DNA methylation, the detection of specific somatic mutations, and *IgH* gene rearrangement. Although the gold standard method for molecular profiling in lymphomas is based on lymph node or tissue biopsies, they have limitations. Lymphomas are not always expressed in peripheral blood, making it difficult to use the concept of the "minimal residual disease" when employing conventional methods. The use of “liquid biopsies” has obstacles in lymphomas, where it is difficult to identify targets applicable to the diversity of these neoplasms. For example, *IGH-BCL2* translocations may be detectable in the blood of patients with FL, transformed FL, and a subset of DLBCL, but there is a variability in the breakpoint regions [92].

#### 4.2.1. Diffuse Large B-Cell Lymphoma

Diffuse large B-cell lymphoma (DLBCL) is the most common type of newly diagnosed lymphoma cases. Three major molecular subtype categories of DLBCL have been defined: the activated B-cell-like (ABC) subtype, the germinal center B-cell-like (GCB) subtype, and the primary mediastinal BCL (PMBL) subtype.

Normal and tumor B-cells express the antigen receptors of B-lymphocytes (BCR), composed of paired heavy and light immunoglobulin chains; there is a wide diversity in the characteristics of the molecule that results from the random rearrangement of the variable diversity joining (VDJ) in the progenitor B-cells, producing unique clonotypes in each lymphoma, which may be useful in anticipating relapses in the peripheral blood, if the relapse comes from the diagnosis clone. However, this technique has disadvantages: There are some DLBCLs with unproductive VDJ rearrangements.

DLBCL is an aggressive and heterogeneous lymphoma, and studies have shown that cfDNA may have a position in the diagnosis of the disease. Various reports have shown a higher amount of cfDNA in patients with this lymphoma, compared to healthy patients, as well as a decrease in the amount of cfDNA when the treatment is effective, and a correlation between the quantification of cfDNA and the adverse prognosis of the disease [21,93,94,95,96].

Digital PCR (dPCR) has been used for detecting common mutations in DLBCL, such as *XPO1 EF71K*, *EZH2 Y641*, and *MYD88 L265P*. Roschewski et al. established the pertinence and the prognostic impact of NGS-based molecular circulating tumor DNA monitoring in a retrospective cohort of 126 patients. The detectable ctDNA in the surveillance serum samples was a strong predictor for disease progression (with an HR of 228) [95,97,98,99,100,101].

Li et al. studied APP gene mutations using RT-PCR in 174 patients (98 DLBCL patients) and concluded that an increase in the cfDNA was associated with the advanced stages of the disease, elevated LDH levels, and a higher prognosis score, with an inferior two-year PFS in patients with high levels of cfDNA [102].

In another study, presented by Rossi et al., a rapid clearance of mutations from cfDNA, among the responsive patients with DLBCL treated with R-CHOP, was demonstrated using CAncer Personalized Profiling by deep Sequencing (CAPPseq) technology, as well as the persistence of basal mutations in the plasma cfDNA of refractory patients [103].

Kurtz et al. studied the dynamics of ctDNA from 217 patients with DLBCL, using CAPPseq, and assessed the prognostic value of ctDNA, regarding risk factors. This study demonstrated that pretreatment ctDNA levels and molecular responses are independently prognostic of outcomes in DLBCL. A two-log drop in ctDNA levels after two chemotherapy courses was associated with an eventual complete response and cure [104]. In another study, Rivas-Delgado et al. reported that high ctDNA levels were associated with a lower response [105,106].

Some studies have demonstrated that ctDNA can be an indicator of the MRD before the lymphoma relapse in DLBCL or T-cell lymphoblastic lymphoma [99,107,108,109]. Shin et al. designed a panel of 66 genes associated with NHLs and analyzed the plasma cfDNA from patients with various subtypes using NGS; this study showed that the level of ctDNA was decreased in patients with a response to therapy, and it increased in patients with disease progression [110]. However, we must be cautious, because some reports have shown that some mutations, such as *TP53* or *DNMT3A*, could have their origin in clonal hematopoiesis of an indeterminate potential (CHIP) [96,111].

Monitoring the MRD with ctDNA may be helpful, following CAR-T-cell therapy [112,113].

#### 4.2.2. Mantle Cell Lymphoma

Mantle cell lymphoma (MCL) is an aggressive lymphoma with a survival median of more than five years. Its management is variable, depending on the risk and patient performance status, and includes chemotherapy, immunotherapy, or autologous stem cell transplantation. The MRD is not very standardized, although some studies have been conducted using the PCR amplification of *IgH* rearrangements, demonstrating an impact on the clinical outcomes of the patient after autologous transplantation [114,115,116]. Lakhotia et al. used NGS to detect Ig heavy and light chains, and *CCND1* and *BCL2* gene mutations in ctDNA; they concluded that the baseline ctDNA correlated with the total metabolic tumor volume on the PET scans, and the clearance of ctDNA after one cycle of DA-EPOCH-R + BZ was strongly associated with a superior median PFS (76.4 vs. 20.7 months, *p* = 0.0037) and a trend toward a superior four-year overall survival (OS) (92.3% vs. 73.0%, *p* = 0.23). The clearance of ctDNA after two cycles of DA-EPOCH-R + BZ was also associated with a superior median progression-free survival (PFS) (32.4 vs. 21.4 months, *p* = 0.015) and a trend toward a superior median OS (82.2 vs. 73.2 months, *p* = 0.15) [117]. Agarwal et al. reported that ctDNA provides valuable prognostic information and enables the real-time assessment of tumor evolution [118].

#### 4.2.3. Follicular Lymphoma

Follicular lymphoma (FL) is the most common indolent lymphoma, and it usually has a prolonged survival (more than 10 years), but the prognosis is variable; treatment consists of immunotherapy (anti CD-20) with or without chemotherapy [119].

Sarkozy et al. reported a study with 133 patients using NGS to detect *VDJ* mutations, showing that a high ctDNA level is associated with a worse PFS (HR = 6.2, 95% CI 2–162, *p* = 0.001) (*p* = 0.14, 0.52, and 0.25 for FLIPI, bone marrow involvement, and the presence of circulating lymphoma cells, respectively) [120].

Delfau-Larue et al. tried to correlate CTCs, cfDNA, and the total metabolic tumor volume (TMTV) with patients’ outcomes using ddPCR and the *IgH* rearrangements. They reported that CTCs and cfDNA were correlated with TMTV, and that the four-year PFS was lower in patients with a TMTV > 510 cm3 (*p* = 0.0004), CTCs > 0.0018 PB cells (*p* = 0.03), and cfDNA > 2550 equivalent-genome/mL (*p* = 0.04) [121]. Another study using NGS in 29 patients with FL showed that the MRD positivity in the interim, or at the end of treatment, results in significantly inferior PFS (median 12 months vs. not reached, *p* = 0.009) [122].

#### 4.2.4. Primary Central Nervous System Lymphoma

Primary central nervous system lymphoma (PCNSL) is an aggressive and infrequent lymphoma [123]. In this lymphoma, the *MYD88 L265P* mutation has been studied in cfDNA using PCR and targeted deep sequencing (TDS). Hattori et al. found this mutation in diagnostics in the majority of patients, but, after chemotherapy, the mutation was undetectable; Yoon et al. concluded that using ctDNA has a limited value for predicting treatment outcomes because it has a low detection efficiency [124,125]. Moreover, for the diagnosis, it may be useful to detect the *MYD88 L265P* mutation in the cfDNA from cerebrospinal fluid [126,127,128].

#### 4.2.5. T-Cell Lymphomas

In T-cell lymphomas, which is less common than B-cell lymphomas, there is less evidence for the use of liquid biopsy. For the diagnosis, and to determine the mutational landscape, as in an angioimmunoblastic T-cell lymphoma, the detection of *HOAG17V/IDH2R172* mutations primarily takes place using an allele-specific polymerase chain reaction (AS-PCR). However, in a previous study with 20 patients, it was not possible to demonstrate an association between mutations and the clinical parameters or survival [129]. Milkjovic et al. used dPCR to detect TCR rearrangements in 34 patients with peripheral T-cell lymphomas (PTCL), and reported a median 2.6-log decrease in their ctDNA levels after the first two cycles of treatment, as well as the early clearance of ctDNA after cycle 2, were not associated with a statistically significant improvement in EFS (median (95% CI), 8.4 (0.1–NR) vs. 2.0 (0.1–NR) years; *p* = 0.32) or OS (median, 8.4 (0.3–NR) vs. 7.0 (0.5–NR) years; *p* = 0.44) [130].

#### 4.2.6. Hodgkin Lymphoma

Hodgkin lymphoma (HL) represents 10% of newly diagnosed lymphomas; the standard initial treatment is chemotherapy and/or radiotherapy, with a five-year progression-free survival rate of 65–90% [89,131]. Studies have been conducted in the attempt to detect peripheral blood mutations in cfDNA using NGS, finding that *XPO1 E571K, ATM, KMT2D*, and *TP53* are frequently mutated in HL [132,133].

In 2018, Spina et al. clarified the genetic landscape of classic HL using CAPP-seq on cfDNA, establishing that *STAT6, TNFAIP3, ITPKB, GNA13, B2M, ATM, SPEN*, and *XPO1* are the most commonly mutated genes [134].

Regarding the MRD, Camus et al. reported that the *XPO1 E571K* mutation may be used as a biomarker in classical HL, using digital PCR, because in their study, patients with a detectable *XPO1* mutation at the end of treatment could have a shorter free progression survival, but the results were not statistically significant [135].

Additionally, Spina et al. reported that a two-log reduction in cfDNA, or greater, between the diagnosis and after the two chemotherapy courses, was linked to a complete metabolic response and cure. However, these studies have several limitations, and more evidence is needed to incorporate liquid biopsies into the monitoring of the MRD in lymphomas [130,134].

#### 4.2.7. Chronic Lymphocytic Leukemia

Chronic lymphocytic leukemia (CLL) is the most common leukemia in the Western world, and it has a variable clinical outcome [136]. In CLL, the MRD is not as well implemented as in other hematologic malignancies, but more and more studies reflect its importance in achieving better progression-free and overall survival rates [137,138,139]. The most widely used techniques are MFC and RQ-PCR. Although this is a circulating disease in the peripheral blood, some studies have demonstrated the usefulness of ctDNA, using digital PCR and targeted sequencing, in obtaining a wide mutational landscape of the disease, or when the disease is localized in an organ, as well as to assess the clonal evolution. More studies are needed to implement the MRD monitoring as a complementary method to those currently used [140].

**Table 7 cancers-14-01310-t007:** Liquid biopsies in lymphomas and chronic lymphocytic leukemia.

Target	Methods	Cohort Size/Disease Stage	Evidence: Key Points	Application	Ref.
*β-globin* gene	qPCR	*n* = 142 DLBCL, *n* = 63 FL, *n* = 24 MCL, *n* = 10 HL, *n* = 45	Increased levels of plasma DNA were associated with advanced stages of disease, B symptoms, elevated LDH levels, and age >60 years In HL, histological signs of necrosis and grade 2 nodular sclerosis were associated with increased plasma DNA Elevated plasma DNA levels were associated with an inferior failure-free survival in patients with HL (*p* = 0.01) and DLBCL (*p* = 0.03)	Concordance Prognosis	Hohaus et al., 2009 [21]
*IgH* gene rearrangements	Locus-specific primer sets por IgH and IgK	*n* = 17 DLBCL, *n* = 15 MLBCL, *n* = 2 Diagnosis and post-treatment	No correlation between the level of ctDNA and clinical characteristics No correlation between the level of ctDNA and the LDH level	Response assessment Prognosis	Armand et al., 2013 [97]
*IgH/TCR* rearrangements	RQ-PCRNGS	*n* = 68 B-NHL, *n* = 37 T-NHL, *n* = 10 HL, *n* = 5 CLL, *n* = 16 Pre- and post-HSCT	Detectable ctDNA three months after HSCT had inferior PFS (*p* = 0.033) and an increased risk of relapse/progression (*p* = 0.0006) 89% of patients with relapse or progression had detectable ctDNA prior to, or at the time of, progression	Prognosis MRD	Herrera et al., 2016 [100]
Somatic gene	Targeted NGS	*n* = 32 B-NHL, *n* = 18 NK or T-NHL, *n* = 9 Other, *n* = 5	ctDNA, at the time around CR/PR, displayed a dramatic decrease in the VAF and number of variants PD displayed a significant elevation of the ctDNA level in VAF	Concordance	Shin et al., 2019 [110]
*APP* gene	RT-PCR	*n* = 174 DLBCL, *n* = 98 HL, *n* = 18 TCL, *n* = 9 NK/T cell lymphoma, *n* = 21 Other B-NHL, *n* = 28 Diagnosis	Lymphoma patients had a higher mean level of cfDNA compared to healthy donors (*p* > 0.0001) Increase in cfDNA was associated with an advanced stage of disease, elevated LDH levels, and a higher prognosis score In patients with DLBCL, high levels of cfDNA showed an inferior two-year PFS	Prognosis	Li et al., 2017 [102]
*IgH* gene rearrangements	NGS (ctDNA) and PCR vs. CT	DLBCL, *n* = 126 Diagnosis and post-treatment	ctDNA may be a sensitive and specific measure of disease Detectable ctDNA means an HR of 228 for clinical progression Patients with early stages (1 or 2) of disease have lower ctDNA, and their LDH level correlates with ctDNA level	Response assessment Staging	Roschewski et al., 2015 [99]
*IgH* gene rearrangements	Ig-HTS vs. PET/CT	DLBCL, *n* = 75 Diagnosis or recurrence	cfDNA correlates with tumor burden measured by PET/CT (*p* = 0.002) cfDNA correlates with PET/CT better than circulating leukocytes cfDNA was better detected in relapse (*p* = 0.001) and precedes PET/CT detection of relapse (*p* < 0.0001)	Surveillance after complete remission	Kurtz et al., 2015 [98]
Somatic mutations	CAPP-seq Ultra-deep targeted NGS	DLBCL, *n* = 30 Diagnosis and post-treatment	Rapid clearance of DLBCL mutations in the cfDNA of responding patients	Response assessment	Rossi et al., 2017 [103]
*V(D)J* rearrangements	NGS	DLBCL, *n* = 6 Pre- and post-CAR-T-cell therapy	MRD by ctDNA correlated with clinical and radiological outcomes for all patients at day 28+ Increasing ctDNA temporally preceded PD in a majority of patients (4/5), and all patients (5/5) had increasing ctDNA at the time of PET-CT-confirmed PD The calculated MTV pre-CAR, and on day 28, showed a strong correlation	MRD Concordance	Hossain et al., 2019 [112]
*L1PA2*	qPCR	DLBCL, *n* = 40 (and 38 controls) Diagnosis	Higher cfDNA in DLBCL patients than in control patients cfDNA level showed an association with >60 years, B symptoms, IPI score, and different disease staging The elevated concentrations of plasma cfDNA correlated with OS (*p* = 0.022)	Prognosis	Eskandari et al., 2019 [94]
Somatic gene	CAPP-seq	DLBCL, *n* = 217 Diagnosis, relapse, or recurrence	Pretreatment levels were prognostic in front-line and salvage settings Patients receiving front-line therapy, achieving EMR (with a 2-log decrease after one cycle) or MMR (a 2.5-log decrease after two cycles), had superior outcomes at 24 months	Prognosis	Kurtz et al., 2018 [104]
Somatic gene	Targeted NGS	DLBCL, *n* = 79 Diagnosis and post-treatment	A higher amount of ctDNA significantly correlated with tumor burden (clinical parameters and MTV) High ctDNA levels (>2.5 log hGE/mL) were associated with lower CR (65% vs. 96%, *p* < 0.004), shorter PFS (65% vs. 85%, *p* = 0.038), and OS at two years (73% vs. 100%, *p* = 0.007)	Concordance Prognosis	Rivas-Delgado et al., 2021 [105]
Ig heavy and light chains and *CCND1* and *BCL2* genes	NGS	MCL, *n* = 53 Diagnosis and post-treatment	Baseline ctDNA correlated with total metabolic tumor volume on PET scan (*r_s_* = 0.74) but was not associated with PFS (*p* = 0.45) or OS (*p* = 0.22) Clearance of ctDNA after one cycle of DA-EPOCH-R + BZ was strongly associated with a superior median PFS (76.4 vs. 20.7 months, *p* = 0.0037) and a trend toward superior four-year OS (92.3% vs. 73.0%, *p* = 0.23) Clearance of ctDNA after two cycles of DA-EPOCH-R + BZ was also associated with a superior median PFS (32.4 vs. 21.4 months, *p* = 0.015) and a trend toward superior median OS (82.2 vs. 73.2 months, *p* = 0.15)	Concordance Prognosis	Lakhotia et al., 2018 [117]
*V(D)J* gene	NGS	FL, *n* = 133 Diagnosis and post-treatment	High ctDNA level was associated with a worse PFS (HR = 6.2, 95% CI 2–162, *p* = 0.001) (*p* = 0.14, 0.52, and 0.25 for FLIPI, bone marrow involvement, and presence of circulating lymphoma cells, respectively) Four patients with a high ctDNA level had a median PFS of only 9.8 months versus those not reached for the 12 patients with a low ctDNA level (*p* = 0.002)	Concordance Prognosis	Sarkozy et al., 2017 [120]
*IgH* gene rearrangements	ddPCR	FL, *n* = 133 Diagnosis and post-treatment	Significant correlation between TMTV and both CTCs (*p* < 0.0001) and cfDNA (*p* < 0.0001) Four-year PFS was lower in patients with TMTV > 510 cm^3^ (*p* = 0.0004), CTCs > 0.0018 PB cells (*p* = 0.03), or cfDNA > 2550 equivalent-genome/mL (*p* = 0.04)	Concordance Prognosis	Delfau-Larue et al., 2018 [121]
Somatic mutations	NGS	FL, *n* = 27 Diagnosis and post-treatment	MRD positivity in the interim or at the end of treatment resulted in significantly inferior PFS (median 12 months vs. not reached, *p* = 0.009)	Concordance Follow-up	Jimenez-Ubieto et al., 2020 [122]
*MYD88* gene	ddPCR TGS	PCNSL, *n* = 14 Diagnosis and post-treatment	MYD88 p.L265P mutation was found in tumor-derived DNA from all 14 patients (14/14, 100%) Among 14 cell-free DNAs evaluated by ddPCR (14/14) and TDS (13/14), the *MYD88 L265P* mutation was detected in eight out of 14 (ddPCR) and in 0 out of 13 (TDS) samples, respectively After chemotherapy, the *MYD88 L265P* mutation in cell-free DNA was traced to five patients; the mutations disappeared after chemotherapy, and they remained undetectable in all patients	Concordance Follow-up	Hattori et al., 2018 [124]
*MYD88* gene	ddPCR	PCNSL, *n* = 29 Diagnosis	MYD288 p.(L265P) was detected in 73% CSF cfDNA samples and 40% of plasma samples	Concordance	Hiemcke-Jiwa et al., 2019 [126]
*MYD88* gene	ddPCR	PCNSL, *n* = 11 Diagnosis and relapse	*MYD288 p.(L265P)* was detected in 86% of the CSF samples	Concordance	Rimelen et al., 2019 [127]
*MYD88* gene	ddPCR	PCNSL, *n* = 42 Diagnosis	*MYD88* mutation status was successfully determined in 28 CSF cfDNA samples (66.7%)	Concordance	Yamagishi et al., 2021 [128]
*RHOAG17V* and *IDH2R172* mutations	AS-PCR	PCNSL, *n* = 20 Diagnosis	14 (70%) and 3 (15%) of the 20 patients generated AS-PCR products indicative of the presence of *RHOAG17V* and *IDH2R172* There was no association between *RHOAG17V*/*IDH2R172* mutations and clinical parameters or survival	Concordance Response assessment	Hayashida et al., 2020 [129]
*TCR* rearrangements	dPCR	PTCLs, *n* = 34: ALCL, *n* = 10 PTCL-NOS, *n* = 10 Other, *n* = 14 Diagnosis	Median 2.6-log decrease in the ctDNA level after the first two cycles of treatment Early clearance of ctDNA after cycle 2 was not associated with a statistically significant improvement in EFS (median (95% CI), 8.4 (0.1–NR) vs. 2.0 (0.1-NR) years; *p* = 0.32) or OS (median, 8.4 (0.3–NR) vs. 7.0 (0.5–NR) years; *p* = 0.44) In six (75%) of the progressors, ctDNA was positive before the detection of clinical relapse	Response assessment	Miljkovic et al., 2021 [130]
*XPO1* gene	dPCR	cHL, *n* = 94 Diagnosis and post-treatment	Concordance of the *XPO1 E571K* mutation dPCR results within the 50 biopsy/plasma DNA pairs was highly significant (*p* = 0.0179) Patients with a detectable *XPO1* mutation at the end of treatment displayed a trend toward shorter two-year PFS, compared to patients with undetectable mutations in plasma cell-free DNA (2-year PFS = 57.1%, 95% CI 30.1–100% versus two-year PFS = 90.5%, 95% CI 78.8–100%, respectively, *p* = 0.0601)	Concordance Response assessment Prognosis MRD	Camus et al., 2016 [135]
*STAT6* mutations	NGSCAPP-seq	cHl, *n* = 112 Newly diagnosed, *n* = 80 Refractory, *n* = 32	A 2-log drop in ctDNA after two chemotherapy courses was associated with a complete response and cure A drop of less than 2-log in ctDNA after two ABVD courses was associated with progression and inferior survival ctDNA quantification after two chemotherapy courses may have prognostic implications, and ctDNA may complement interim PET/CT in informing on patients’ outcomes	Response assessment Prognosis	Spina et al., 2018 [134]
*NFKBIE*, *TNFAIP3*, *STAT6*, *PTPN1*, *B2M*, *XPO1*, *ITPKB*, *GNA13*, and *SOCS1* genes	NGS	HL, *n* = 60 Diagnosis and post-treatment	ctDNA concentration and genotype were correlated with clinical characteristics and presentation No statistically significant difference between the concentration of cfDNA (ng/mL of plasma) after C2 among DS 1–3 patients (35 patients, median 35 ng/mL (range: 20.4–260) versus DS 4–5 patients (seven patients, median 36.2 ng/mL (range: 21.8–80), *p* = 0.79)	Concordance Prognosis	Camus et al., 2021 [133]

Ref., reference; RQ-PCR, real-time quantitative polymerase chain reaction; MRD, minimal residual disease; PB, peripheral blood; cfDNA, cell-/non-Hodgkin lymphoma; HL, Hodgkin lymphoma; DLBCL, diffuse large B -cell lymphoma; PTCL-NOS: peripheral T-cell lymphoma-not otherwise specified; qPCR, quantitative PCR; FL, follicular lymphoma; MCL, mantle cell lymphoma; LDH, lactate dehydrogenase; ctDNA, circulant tumor DNA; MLBCL, mediastinal large B-cell lymphoma; HSCT, hematopoietic stem cell transplantation; PFS, progression-free survival; NGS, next-generation sequencing; VAF, variant allele frequency; PD, progressive disease; CT, computerized tomography; PET, positron-emission tomography; CAPP-seq, CAncer Personalized Profiling by deep Sequencing; CAR, chimeric antigen receptor; OS, overall survival; MTV, metabolic tumor volume; EMR, early molecular response; MMR, major molecular response; HR, hazard ratio; ddPCR, droplet digital PCR; FLIPI, Follicular Lymphoma International Prognostic Index; TMTV, total MTV; PCNSL, primary central nervous system lymphoma; TDS, target deep sequencing; CSF, cerebrospinal fluid; AS-PCR, allele-specific PCR; dPCR, digital PCR; DS, Deauville score.

### 4.3. Multiple Myeloma 

Multiple myeloma (MM) is a clonal plasma cell proliferative disorder characterized by bone lesions, whose diagnosis is based on the presence of clinical, biochemical, histopathological, and radiological markers of disease. The appearance of new drugs and the improvement of supportive treatment has produced an increase in the median survival, which is currently estimated at five years in developed countries [141,142].

The MRD in MM has been shown to be a predictive factor for PFS and OS; it is performed by multiparametric flow cytometry (MFC) or by allele-specific oligonucleotide PCR (ASO-PCR) sequencing in bone marrow, with a sensitivity of 10^−5^–10^−6^ [143,144,145,146], or by NGS, to detect *IgH* rearrangements with a sensitivity of 10^−6^ [147].

MM is a disease of complex molecular biology, with different clones of malignant plasma cells in the same patient; these clones evolve in the natural history of the disease. Several genes are involved in many patients, such as *KRAS*, *NRAS*, or *BRAF* [148,149]. In MM, circulating tumor cells are released from the primary tumor into the bloodstream, usually reaching another location in the bone marrow; this process is early on in the carcinogenesis. Different groups have studied circulating tumor cells in MM and other plasma cell dyscrasias; they can be particularly useful in extramedullary, oligosecretory, and non-secretory disease settings [150,151,152,153,154,155].

As in other hematologic neoplasms, circulating tumor DNA can be analyzed in MM through tumor-specific mutations or genetic aberrations. In recent years, the number of articles on this subject has increased (Table 8). In 2015, Sata et al. evaluated the tumor burden in mRNA from peripheral blood cells, whole bone marrow cells, the CD20+CD38L B-cell population in bone marrow, and the cell-free DNA from the sera of patients with MM, and compared them using ASO-PCR. This study, with 30 patients, found statistically significant correlations between the ASO-PCR levels in bone marrow cells and the peripheral blood cells, which suggests that clonogenic plasma cells or MM precursor cells may circulate in peripheral blood, but the ASO-PCR values in the cell-free DNA from the sera did not correlate with those in either the bone marrow or the peripheral blood cells [156].

Oberle et al. conducted a study in 27 MM patients to explore the clonotypic V(D)J rearrangement for monitoring circulating myeloma cells and cell-free myeloma DNA. The positivity for circulating myeloma cells/cell-free myeloma was associated with a conventional remission status, and the majority of non-responders/progressors had evidence of persistent circulating myeloma cells/cell-free myeloma DNA. However, the positivity for circulating myeloma cells and for cell-free myeloma DNA were discordant in 30% of cases, which indicates that cell-free myeloma DNA may not be generated entirely by circulating myeloma cells and may reflect the overall tumor burden [157].

Mithraprabhu et al. analyzed the plasma-derived circulating free tumor DNA as an adjunct to bone marrow biopsies for mutational characterization (for *KRAS*, *NRAS*, *BRAF*, and *TP53*) and for tracking disease progression in 33 relapsed/refractory and 15 newly diagnosed MM patients, in comparison to 12 healthy donors by NGS, showing a higher amount of cfDNA in MM patients. Some mutations were found only in the cfDNA. Although there were few patients in the study, it may confirm the spatial heterogeneity of MM [158].

Rustad et al. explored the presence of circulating tumor DNA, monitoring recurrent mutations (*NRAS*, *KRAS*, and *BRAF*) using ddPCR, and comparing it to bone marrow plasma cells. They observed a correlation between the concentration of mutated alleles in the serum and the fraction in bone marrow plasma cells, which may reflect mutated cells, the total tumor mass, and the transformation to a more aggressive disease that is complementary to the M protein [159].

Gerber et al. studied the cfDNA in 28 patients, identifying the clonotypic V(D)J rearrangement as a marker, or genotyping a limited set of genes, and designed a panel of NGS, comparing the characteristics of these patients to samples of bone marrow aspirates from patients with plasma cell disorders in different stages. The amount of cfDNA correlated with some parameters that may indicate the tumor burden, such as the percentage of plasma cell infiltration of bone marrow [160].

Biancon et al. analyzed the disease evolution of 25 MM patients receiving second-line therapy, and the study showed that the levels of *IgH* detected in cfDNA reflected the number of PCs enumerated by MFC, which correlated with clinical outcomes [161].

Other studies have shown that cell-free tumor DNA can be used to obtain the molecular profile of myelomas instead of the bone marrow aspirate, and they have supported the concept of cell-free DNA as a prognostic marker [162,163,164]. In a recent study, after sorting 77 MM patients according to their molecular risk, Deshpande et al. found that cfDNA was higher in high-risk MM, and high cfDNA levels were associated with a worse PFS and OS [165]. In other study, Manzoni et al. analyzed 104 samples from 65 patients (15 monoclonal gammopathy of undetermined significance-MGUS; 33 smoldering multiple myeloma-SMM; and 17 MM) using ultra-deep NGS, concluding that a lower tumor mass correlates with a lower cfDNA tumor fraction, and it could be useful to evaluate MGUS and SMM in the future [166].

Regarding the study of the MRD with cfDNA, there are few published studies that compare it to the currently standardized methods (flow cytometry and DNA sequencing in bone marrow) [167]. Mazzotti et al. demonstrated the absence of a correlation between ctDNA and bone marrow for the MRD by NGS, using only *IgH* gene rearrangements, although only 37 patients were included [168]. Long et al. analyzed 22 plasma samples from eight extramedullary multiple myelomas (EMM) and 23 plasma samples from 10 MM patients without extramedullary spread, with higher cfDNA concentrations in patients with extramedullary spread. After designing sequencing panels targeting the coding sequence regions of the same 22 recurrently mutated genes, 17 different were detected. The authors concluded that cfDNA can be used to track extramedullary disease progression, including the MRD, when plasmacytomas are inaccessible [169].

In other studies, a significant correlation of the quantity of tumor-specific cell-free DNA levels with clinically meaningful events has been found, but the results in the case of the MRD monitoring were not significant [163,170]. The target sequencing of cfDNA cannot, today, achieve the sensitivity of the MRD detection of flow cytometry or PCR; however, in the future, the MRD detection in cell-free DNA may increase its sensitivity combining parameters (patient-specific mutation panels, methylation patterns, copy number alterations, or *IgH* rearrangements), which may add further accuracy to progression-free survival prediction and the detection of the false-negative MRD [171,172,173]. At the present time, the isolated use of cfDNA has no clinical applicability in the study of the MRD in multiple myeloma, although there is increasing evidence, which may initially make it a complementary test for the follow-up of the disease and, in the future, with its improvement, into a fundamental tool for the evaluation of the MRD [155].

**Table 8 cancers-14-01310-t008:** Liquid biopsies in multiple myeloma.

Target	Methods	Cohort Size/Disease Stage	Evidence: Key Points	Application	Ref.
*IgH* gene	ASO-PCR	*n* = 30 Diagnosis and post-treatment	Myeloma cell-derived *IgH* DNA fragments in the sera stayed at similar levels and sometimes increased during treatment	Concordance	Sata et al., 2015 [156]
*V(D)J* rearrangement	NGS	*n* = 27 Diagnosis and post-treatment	At the follow-up, cfm-*V(D)J* in 34% of samples Clear associations were observed between poor remission status and evidence of cfm-V(D)J (regression coefficient 1.49; *p* = 0.001)	Concordance Prognosis	Oberle et al., 2017 [157]
*KRAS*, *NRAS*, *BRAF*, and *TP53* mutations	*ddPCR*	*n* = 60 New diagnosis, *n* = 15 Relapse/refractory, *n* = 33 Diagnosis and post-treatment Normal volunteers, *n* = 12	ctDNA analysis in seven patients revealed an increase in the AF of specific mutant clones coincident with clinical relapse or a potential noninvasive monitoring of MM disease progression	Concordance Response assessment	Mithraprabhu et al., 2017 [158]
*NRAS*, *KRAS*, and *BRAF* mutations	ddPCR	*n* = 18 Diagnosis and post-treatment	12/14 mutated clones were detectable in the serum at each relapse and covaried with M protein	Concordance Response assessment	Rustad et al., 2017 [159]
*KRAS*, *NRAS*, *BRAF*, *EGFR*, and *PIK3CA* mutations	ddPCR	*n* = 53 New diagnosis, *n* = 11 Relapsed, *n* = 42	Higher cfDNA concentrations in MM cohort compared to 56 patients with advanced solid tumors (*p* < 0.001) Concentrations of cfDNA correlated with advanced disease (late relapse compared to early relapse; *p* = 0.016) Mutant AFs were highly concordant between cfDNA and BM (*R*^2^ range, 0.913–0.997)	Concordance Prognosis	Kis et al., 2017 [162]
Somatic mutations	CAPP-seq	*n* = 28 New diagnosis, *n* = 25 Relapsed/refractory, *n* = 3 Diagnosis	The amount of cfDNA correlated with clinical–pathological parameters reflecting tumor load/extension, including BM PC infiltration (*r_s_* = 0.42, *p* = 0.02) Variant allele frequencies in the plasma samples correlated with those in tumor biopsies (*r_s_* = 0.58, *p* = 9.6 × 10^−5^)	Concordance	Gerber et al., 2018 [160]
*IgH* gene rearrangements	ddPCR	*n* = 25 At first relapse	Patients with levels 4.7% (*n* = 12) of the tumor-associated *IgH* sequence before therapy had significantly inferior PFS (median values, 268 vs. 990 days; HR = 3.507, *p* = 0.04988, log rank test) High level of correlation between cfDNA NGS and MFC data (*r* = 0.5831, *p* = 0.0044, Pearson’s correlation test)	Concordance Prognosis	Biancon et al., 2018 [161]
*IgH* gene rearrangements	NGS	*n* = 37 Post-treatment	Minimal correlation between myeloma ctDNA detection at the time of MRD in the BM and quantity of analyzed cell-free DNA (*r* = 0.46; *p* = 0.001)	MRD	Mazzotti et al., 2018 [168]
Somatic mutations	WES	*n* = 163 cfDNA, *n* = 107 CTCs, *n* = 56 Diagnosis	Concordance in clonal somatic mutations (~99%) and copy number alterations (~81%) between liquid and tumor biopsies	Concordance	Manier et al., 2018 [170]
Somatic mutations	WES	*n* = 105 MM patients, *n* = 93 Healthy patients, *n* = 12 Diagnosis	90.5% of all CNV segments in the BM were concordant with cfDNA, whereas 9.5% were discordant	Concordance	Guo et al., 2018 [171]
Somatic mutations	Ultra-deep NGS	*n* = 65 MGUS, *n* = 15 SMM, *n* = 33 MM, *n* = 17 Diagnosis	cfDNA concentrations were significantly lower in MGUS and SMM. On average, they were 2.8-fold lower than in MM (*p* = 0.02)	Concordance Prognosis	Manzoni et al., 2020 [166]
Somatic mutations	NGS	*n* =18 EMM patients, *n* = 8 MM without EM spread, *n* = 10 Diagnosis	ctDNA exhibited strong concordance with time-matched extramedullary plasmacytoma biopsies (*p* = 8.66 × 10^−7^)	Concordance	Long et al., 2020 [169]
Somatic mutations	Targeted NGS	*n* = 77 Newly diagnosed, *n* = 52 Relapsed, *n* = 2 Previously treated, *n* = 23	Weak correlation between ISS and cfDNA levels (*r* = 0.32, *p* = 0.005) A weak correlation was seen with the cfDNA concentration and LDH levels (*r* = 0.44, *p* < 0.0001) cfDNA levels correlated weakly with serum β2m (*r* = 0.33, *p* = 0.003) Correlation between GEP70 risk score with cfDNA levels (*r* = 0.28; *p* = 0.01)	Concordance Prognosis	Deshpande et al., 2021 [165]

Ref., reference; ASO-PCR, allele-specific oligonucleotide PCR; NGS, next-generation sequencing; ddPCR, droplet digital PCR; ctDNA, circulant tumor DNA; AF, allele frequency; MM, multiple myeloma; cfDNA, cell-free DNA; BM, bone marrow; CAPPseq, CAncer Personalized Profiling by deep Sequencing; PC, plasmatic cell; HR, hazard ratio; MFC, multiparameter flow cytometry; WES, whole-exome sequencing; CTC, circulant tumor cell; CNV, copy number variation; MGUS, monoclonal gammopathy of undetermined significance; SMM, smoldering multiple myeloma; EM, extramodular; ISS, International Staging System; LDH, lactate dehydrogenase.

## 5. Current Situation and Preanalytical Recommendations 

### 5.1. Current Situation

In the field of hematology, studies of liquid biopsies, in terms of CTC, have been carried out over the last few years, such as in CML (with a treatment adjustment in terms of *BCR/ABL1* peripheral blood leukocyte RNA in CML patients, as well as adjustments in terms of the *NPM1*-mutated ratio in peripheral blood leukocyte RNA in AML). However, the application of liquid biopsies, in terms of cfDNA, for treatment adjustment (as a biomarker of the MRD) in myeloid pathology has not been highly developed. However, recent publications have shown its usefulness as a complementary technique to CTC or BM studies in the follow-up of the MRD, and they allow an interesting genomic representation of the different tumor clones.

There are several papers on the applicability of cfDNA as a marker for the MRD in B- or T-cell lymphoid malignancies. In these pathologies, there is a lot of information on the usefulness of this approach to evaluate the MRD, where it could be considered a new pathway for the MRD. In chronic lymphocytic leukemia, the evolution of resistance to treatments in real time could be monitored with ctDNA, and in acute leukemias, it can be helpful in the monitoring of early responses to treatments and the prediction of treatment failure. 

NGS is the preferred method for liquid biopsy studies to detect the MRD in hematologic pathology, given the great molecular heterogeneity of these tumors. The main limiting factor of these sequencing tools is the error rate, which varies between 1% and 0.01% depending on the sequencing conditions, starting material, and the analysis pipeline. Variant detection with high confidence occurs at a fraction below 1% and, therefore, requires sufficient depth of coverage (i.e., the number of sequences that "read" any nucleotide position) in both patient and control samples, as well as the use of bioinformatics analyses and algorithms that allow for the application of the appropriate quality control criteria [31]. In order to optimize this procedure, the main recommendations are as follows.

### 5.2. Preanalytical Recommendations

High-quality cfDNA for liquid biopsies is required to avoid contamination by genomic DNA from white blood cells, and to maintain a sufficient fragment length to allow for the conduction of PCR-based methods. Various preanalytical factors can influence sample quality for ctDNA extraction, such as the centrifugation procedure, or the storage conditions. Then, a blood collection tube, containing a preservative that stabilizes nucleated blood cells for delayed separation, can be used. Recent studies have shown similar ctDNA levels at zero hours, or for up to five days, at room temperature in blood collection tubes with preservative, and a 2 h stability if the blood is drawn in EDTA tubes [174]. A correct centrifugation protocol with the aim of limiting genomic DNA contamination of leukocytes is needed. For cfDNA extraction, the sensitivity of the different assays is driven by a DNA input spin column, and the magnetic bead-based isolation methods are influenced by the size of the DNA fragments. Thus, the selected extraction kit must be adapted to the sample, and the use of the membrane-based method promotes the extraction of a high molecular weight cfDNA fragments (>600 bp), whereas the magnetic bead system yields shorter cfDNA fragments. The subsequent storage should be at −20 or −80 °C for up to nine months with good preservation, without thawing, or with less than three thaws.

## 6. Discussion

NGS techniques are rapidly evolving, and their sensitivity is improving, allowing their application even in early-stage diseases. Various assays are available, and the PCR-based methods require precise knowledge of the expected alteration, as well as allowing for the detection of SNV with a very low allele frequency, a relatively low time frame, and low costs. Alternatively, the NGS-based assays allow for the detection of non-hotspot mutations and other variations, such as CNV, but they require more time than PCR assays, as well as a thorough bioinformatics pipeline for analyses, and expertise in the technical and complex interpretations of the data read-out.

Basic NGS workflows are designed to detect mutations above a 1% minimum mutant allele fraction. In ctDNA, this mutant allele fraction is generally reached in a metastatic disease context with a high tumor load. For early-stage cancers, a lower detection threshold is needed for ctDNA mutation detection, and the basic NGS workflow needs further adaptation.

Targeted NGS methods have, therefore, been developed to screen a large number of potential mutations in tumor biopsies with an elevated sensitivity (e.g., Guardant Health, which employs the pre-sequencing preparation of a digital library of individually tagged cfDNA molecules, combined with the post-sequencing bioinformatic reconstruction to eliminate nearly all false positives; the Foundation ACT, with hybrid capture-based assays; CAPP-Seq, with limit-sequenced regions by capturing recurrently mutated genomic regions; Oncomine Lung cfTNA, with capture-based methods to assess 12 genes that are frequently mutated in lung cancer; Trusight tumor 15 assay, with capture-based methods to assess 15 genes that are frequently mutated in solid tumors; Safe sequencing, with the assignment of a unique identifier to each DNA template molecule to be analyzed, as well as the amplification of each uniquely tagged template; and the Archer Reveal cDNA 28kit, with capture-based methods to assess 28 genes that are frequently mutated in solid tumors) [175]. However, there are no specifically-targeted NGS methods with an elevated sensitivity oriented towards hematological tumors.

## 7. Conclusions

The applications of liquid biopsy methods in evaluating the responses to treatment of onco-hematological pathologies have shown their great utility, and they could be imposed in a relatively short time frame, alone or in combination with other methods (e.g., imaging). Most of the studies carried out to date are case reports or small samples, but the results in these, or in larger studies, are encouraging. However, there are still some limitations to overcome, such as improving the obtainment of a sufficient amount of cfDNA and the sensitivity of the techniques used (generally NGS), as well as obtaining validated results in clinical trials.

## Figures and Tables

**Table 5 cancers-14-01310-t005:** Liquid biopsies in myeloproliferative neoplasms.

Target	Methods	Cohort Size/Disease Stage	Evidence: Key Points	Application	Ref.
Somatic mutations	NGS ddPCR	*n* = 107 patients (33 PV, 56 ET, 14 PMF, and 4 uMPNs)	A significant concordance was observed between the amount of cfDNA and PMF (vs. PV), leukocyte count, LDH, *MPL*-mutated group, and those suffering from a thrombotic event at the time of diagnosis or during follow-up For ET cases treated with IFN, a proportional decrease in the *JAKV617F* VAF was observed in granulocytes and cfDNA	Concordance Response assessment	Garcia-Gisbert, et al., 2021 [78]

Ref., reference; NGS, next-generation sequencing; ddPCR, digital droplet PCR; PV, polycythemia vera; ET, essential thrombocythemia; PMF, primary myelofibrosis; uMPNs, unclassifiable myeloproliferative neoplasms; cfDNA, cell-free DNA; LDH, lactate dehydrogenase; IFN, interferon; VAF, variant allele frequency.

**Table 6 cancers-14-01310-t006:** Liquid biopsies in lymphoblastic acute leukemia.

Target	Methods	Cohort Size/Disease Stage	Evidence: Key Points	Application	Ref.
*IgH*/*TCR* rearrangements	RQ-PCR	Precursor B-ALL, *n* = 62 T-ALL, *n* = 22 Diagnosis and post-treatment	Strong correlation in the MRD levels between BM and PB cfDNA (*r*_s_ = 0.849; *p* < 0.01) in T-ALL, but not in the precursor B-ALL	Concordance Response assessment	van der Velden et al., 2002 [85]
*IgH*/*TCR* rearrangements	RQ-PCR in plasma vs. leucocytes	*n* = 21 (2–16 years) Diagnosis and post-treatment	High concordance (86.7%) between the MRD measurements in plasma and leukocytes High DNA levels in ALL at diagnosis that rapidly decrease after initiating treatment	Concordance	Schwarz et al., 2009 [86]
*IgH*/*TCR* rearrangements	RQ-PCR vs. flow cytometry	*n* = 206 Diagnosis and post-treatment	Poor correlation between the two methods in assessing the MRD level, with *R*= −0.0733 Positive PB MRD (detection threshold 10^−4^) had a 5.8-fold higher risk of relapse (*p* = 0.038)	Response assessment by MRD Prognostication	Cheng et al., 2013 [87]

Ref., reference; ALL, acute lymphoblastic leukemia; RQ-PCR, real-time quantitative polymerase chain reaction; MRD, minimal residual disease; PB, peripheral blood; cfDNA, cell-free DNA; *r*_s_, Spearman’s rank correlation coefficient.

## Data Availability

Not applicable.

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
