# Peer review of "The Minimal Residual Disease Using Liquid Biopsies in Hematological Malignancies"

_cancers, 2022, doi:10.3390/cancers14051310_

Round 1
Reviewer 1 Report
A very comprehensive review on a very important and topical subject. The review has presented thoroughly the studies available on methods and approaches followed to detect liquid biopsy based markers linked to haematological malignancies with an emphasis in MRD. Some minor changes are recommended to present the content in a more coherent way and in particular:
Introduction: Sections 1.1-1.3 can be structured in larger groups to avoid the segmentation of paragraphs and thus the spread of the techniques used. So rather than the use of titles "Liquid biopsy components", "Common techniques in DNA liquid biopsies" and "Minimal residual disease using liquid biopsy", the review can be sectioned with an emphasis on the biomarkers and technologies while emphasising on the application of these in different diagnostic stages, clinical application and monitoring studies.
Sections 2-3 (Myeloid and Lymphoid malignancies) could actually be presented before Section 1, so that the readers understand the types of haematological neoplasms and then investigate the liquid biopsy biomarkers and diagnostic platforms. It might be more educative in terms of sequence of contexts and can build up in terms of the clinical translations.
Section 4 title needs to be replaced with a more generic one as it feels like the beginning of discussion, portraying of the clinical need and the assays that are desired to be developed to evaluate response to treatment. The paragraph "References...relapse" should be moved down as it appears out of context and is usually added towards the end of a meta-analysis review on methods.
Conclusion is rather short, a recommendation could be to connect 4.2 with the paragraph in conclusion, and use it as the recommendation/discussion section, and therefore the authors' view on the future of the methods and assays needed to advance the role of liquid biopsy tests in MRD in haematological malignancies.
Author Response
[General Comment] “A very comprehensive review on a very important and topical subject. The review has presented thoroughly the studies available on methods and approaches followed to detect liquid biopsy based markers linked to haematological malignancies with an emphasis in MRD.”
Response: Thank you very much for your kind comment and for your time reviewing the manuscript.
“Some minor changes are recommended to present the content in a more coherent way and in particular”:
[Minor Comment 1] “Introduction: Sections 1.1-1.3 can be structured in larger groups to avoid the segmentation of paragraphs and thus the spread of the techniques used. So rather than the use of titles "Liquid biopsy components", "Common techniques in DNA liquid biopsies" and "Minimal residual disease using liquid biopsy", the review can be sectioned with an emphasis on the biomarkers and technologies while emphasising on the application of these in different diagnostic stages, clinical application and monitoring studies.”
Response: Thank you very much for your interesting and thoughtful comment. We have removed the subsections of section 1.1, making it easier to read, and integrated section 1.2 at the end of section 1.1. On the other hand, after reflection, and taking into account the structure of the manuscript, we have decided to describe the biomarkers and their usefulness in different stages, clinical application and monitoring studies in each section of "Myeloid and lymphoid malignancies". We understand that your proposal is more suitable for a reader with a laboratory point of view, but we believe that for most clinical readers our option is simpler, but if you consider necessary this change we can carry it out.
[Minor Comment 2] “Sections 2-3 (Myeloid and Lymphoid malignancies) could actually be presented before Section 1, so that the readers understand the types of haematological neoplasms and then investigate the liquid biopsy biomarkers and diagnostic platforms. It might be more educative in terms of sequence of contexts and can build up in terms of the clinical translations.”
Response: Thank you very much for your interesting and valuable comment. We have been respectfully reflecting on this suggestion, but we have decided to keep the "Myeloid and Lymphoid Malignancies" sections after the first section, because we think that if the order is reversed, it can make it difficult to follow the manuscript.
[Minor Comment 3] “Section 4 title needs to be replaced with a more generic one as it feels like the beginning of discussion, portraying of the clinical need and the assays that are desired to be developed to evaluate response to treatment. The paragraph "References...relapse" should be moved down as it appears out of context and is usually added towards the end of a meta-analysis review on methods.”
Response: Thank you very much for your kind comment; we have changed the title of the section ("Current situation and preanalytical recommendations"), and we have moved the paragraph "References...relapse", creating a new section ("Methods") at the beginning of the manuscript, where we think that, considering the organization of the manuscript, is more appropriate.
[Minor Comment 4] “Conclusion is rather short, a recommendation could be to connect 4.2 with the paragraph in conclusion, and use it as the recommendation/discussion section, and therefore the authors' view on the future of the methods and assays needed to advance the role of liquid biopsy tests in MRD in haematological malignancies.”
Response: Thank you very much for your valuable comment; we have eliminated the "Conclusion" section and with point 4.2 we have created a "Discussion" section, connecting former 4.2 with the paragraph in conclusion, which gives more continuity to the exposed arguments.
Reviewer 2 Report
In this review the authors have focused on clinical applications of cfDNA on minimal residual disease in hematological malignancies.
The present manuscript is of significant interest and relevant to therapeutic research in the field of blood malignancies. Although review papers on similar subject matter have been published recently (e.g. Lim, J.K.; Kuss, B.; Talaulikar, D. Role of Cell-Free DNA in Haematological Malignancies. Pathology (Phila.) 2021, 53, 941 416–426, doi:10.1016/j.pathol.2021.01.004.), this one is distinguished by a large number of research studies cited and a comprehensive approach to the topic. This paper should be accepted for publication after some minor reviews.
- Tables 3-8: The last column “reference” should be supplemented with the name of the first author and the year of publication. Furthermore, adding the first column with an ordinary number could give a quick overview on the number of papers on the given topic.
- Page 2 line 78: the abbreviation “ctDNA” has not been explained before
- Page 6 line 231: the abbreviation “VAF” has not been explained before (this is further explained on page 9 line 302)
- Page 7 line 242: the term “DTA gene mutations” has not been explained
- The process of papers selection for this review (page 22 lines 675-687) should be described in the first, not in the last part of the manuscript
- In the discussion/conclusion that most studies on this topic at present are still limited to basic sciences, case reports and small clinical studies should be more emphasized.
Author Response
[General Comment] “In this review the authors have focused on clinical applications of cfDNA on minimal residual disease in hematological malignancies.
The present manuscript is of significant interest and relevant to therapeutic research in the field of blood malignancies. Although review papers on similar subject matter have been published recently (e.g. Lim, J.K.; Kuss, B.; Talaulikar, D. Role of Cell-Free DNA in Haematological Malignancies. Pathology (Phila.) 2021, 53, 941 416–426, doi:10.1016/j.pathol.2021.01.004.), this one is distinguished by a large number of research studies cited and a comprehensive approach to the topic.”
Response: Thank you very much for your comment and for the time you have spent reviewing the manuscript.
[Minor Comment 1] “Tables 3-8: The last column “reference” should be supplemented with the name of the first author and the year of publication. Furthermore, adding the first column with an ordinary number could give a quick overview on the number of papers on the given topic.”
Response: Thank you for your comment. With this contribution, we agree the manuscript is easier to read; also, it is easier to know the number of studies published to date.
[Minor Comment 2] “Page 2 line 78: the abbreviation “ctDNA” has not been explained before”
Response: Thank you very much for the reminder. We have made revision accordingly.
[Minor Comment 3] “Page 6 line 231: the abbreviation “VAF” has not been explained before (this is further explained on page 9 line 302)”
Response: Thank you very much for the reminder. We have made revision accordingly.
[Minor Comment 4] “Page 7 line 242: the term “DTA gene mutations” has not been explained”
Response: Thank you very much for the reminder. We have made revision accordingly.
[Minor Comment 5] “The process of papers selection for this review (page 22 lines 675-687) should be described in the first, not in the last part of the manuscript”
Response: We moved this paragraph and we have created a new section called “Methods”.
[Minor Comment 6] “In the discussion/conclusion that most studies on this topic at present are still limited to basic sciences, case reports and small clinical studies should be more emphasized.”
Response: Thank you for your comment; we have further emphasized at the end of the new "Discussion" section that most of the current evidence is based on small studies.
Reviewer 3 Report
The manuscript "Minimal residual disease using liquid biopsies in hematological malignancies" is an interesting review of a currently "hot topic".
The manuscript is well-organized and is easy to follow, with multiple great tables.
The only addition I would suggest is to add more details to certain methods, such as dPCR and NGS, with describing why their sensitivity is better etc.
I suggest to accept the manuscript with this minor addition.
Author Response
[General Comment] “The manuscript "Minimal residual disease using liquid biopsies in hematological malignancies" is an interesting review of a currently "hot topic".
The manuscript is well-organized and is easy to follow, with multiple great tables.
I suggest to accept the manuscript with this minor addition.”
Response: Thank you very much for your comment and for the time you have spent reviewing the manuscript.
[Minor Comment 1] “The only addition I would suggest is to add more details to certain methods, such as dPCR and NGS, with describing why their sensitivity is better etc.”
Response: Thank you very much for your kind comment; we have described in more detail the techniques and why they are used.